# Generalized Persistence for Equivariant Operators in Machine Learning

**Mattia G. Bergomi [1]**, **Massimo Ferri [2,]***, **Alessandro Mella [3]** and **Pietro Vertechi [4]**

1   Independent Researcher, 20124 Milan, Italy; mattiagbergomi@gmail.com
2   ARCES and Department of Mathematics, University of Bologna, 40126 Bologna, Italy
3   Independent Researcher, 37100 Verona, Italy; alessandro.mella.92@gmail.com
4   Independent Researcher, 34100 Trieste, Italy; pietro.vertechi@protonmail.com
*   Correspondence: massimo.ferri@unibo.it

**Abstract:** Artificial neural networks can learn complex, salient data features to achieve a given task. On the opposite end of the spectrum, mathematically grounded methods such as topological data analysis allow users to design analysis pipelines fully aware of data constraints and symmetries. We introduce an original class of neural network layers based on a generalization of topological persistence. The proposed *persistence-based* layers allow the users to encode specific data properties (e.g., equivariance) easily. Additionally, these layers can be trained through standard optimization procedures (backpropagation) and composed with classical layers. We test the performance of generalized persistence-based layers as pooling operators in convolutional neural networks for image classification on the MNIST, Fashion-MNIST and CIFAR-10 datasets.

**Keywords:** topological persistence; rank-based persistence; pooling; steady persistence





## 1. Introduction

Artificial neural networks (ANNs) can approximate arbitrarily complex functions provided that they are fed with a high-quality, sufficiently large training set. Users do not need to develop a profound knowledge of the data involved in the task. However, they do not control the features that the network will learn to solve the task. The lack of control makes it difficult to predict the generalization capacity of ANN-based analysis pipelines, causing pathologies such as vulnerability to adversarial attacks [1] or requiring the investigation of custom data augmentation algorithms [2] to reduce errors or tame unwanted behaviors caused by noisy features interpreted as salient by the network. On the contrary, the topological persistence (TP) requires the user to explicitly declare—under the form of a continuous real-valued function—which features of the data are relevant to tackle a given task. This procedure does not require harvesting training data and gives the user complete control over the features used to solve the task.

Both the ANN and TP frameworks have apparent drawbacks; gathering the massive training datasets required to train complex neural architectures can be extremely hard. Symmetries of the data can be leveraged to reduce the dimensionality of the parameter space of ANNs and learn efficiently from smaller datasets [3–5]. However, it is often difficult to adapt such constructions to arbitrary features. Instead, in the TP frameworks, accessing sufficient information to determine what features are needed to achieve the desired results can be equally daunting. Moreover, TP is bound to topological data types (e.g., triangulable manifolds). Thus, data lacking such topological structures need to be mapped—often via complex transformations—to topological spaces. See [5–7] for examples of topological constructions mapping graphs to simplicial complexes.

Interactions between TP and deep learning are of broad interest [8–10]. Ideally, combining the two methods would yield constrainable and learnable models composable with

state-of-the-art neural networks. However, we believe that the need to map data to topological objects and express their features as critical points of continuous functions hinders the development of TP-inspired neural layers.

### 1.1. Aim

At the crossroad between artificial neural network and topological persistence, we provide an algorithmic approach to the design of learnable persistence-based layers focusing on generality in terms of data types, usability through a streamlined user interface, and flexibility. We do this by building on the framework of rank-based persistence [11], which allows us to avoid auxiliary topological constructions mapping data to topological objects. This approach broadens the spectrum of applicability of our solutions, naturally including data types such as undirected and directed graphs, and metric spaces. Finally, in the attempt of simplifying the TP's algorithmic pipeline, we leverage the notion of persistent features [12]. Persistent features allow us to define learnable, persistence-based neural network layers based on Boolean features of the data, rather than defining them as continuous functions. Finally, such layers can be easily constrainable with respect to data symmetries.

### 1.2. Contribution

In this study, we define and provide constructive examples of operators based on persistent features and discuss their properties, particularly constrainability and noise robustness. After showcasing these properties on images, we introduce an original neural network layer, namely the *persistent feature-based layer*. We base our proposal on two primary principles. On the one hand, we aim to learn relevant (persistent) features directly from data points. On the other hand, we provide simple strategies to take advantage of locality and equivariance, which are two critical notions for analyzing structured data. We provide an algorithm for a persistence-based pooling operator, test it on several architectures and datasets, and compare it with some classical pooling layers.

### 1.3. Structure

In Section 2, we retrace the interplay between networks and topological persistence, in particular for the use of the latter in a neural network. Section 3 intuitively introduces the central mathematical concepts involved in the definition of a persistence-based layer: persistent homology, rank-based persistence, and persistent feature. In Section 4, we devise an image-filtering algorithm based on persistent features, discuss its main properties, and provide examples. Thereafter, we define a persistence-based neural network layer and specialize it to act as a pooling layer in convolutional neural networks. Computational experiments in Section 5 evaluate and compare the performance of the proposed pooling layer in an image-classification task on several datasets and on two different architectures. In the same section, we provide a qualitative analysis of the most salient features detected by the persistence-based pooling, the classical max-pooling, and LEAP [13].

## 2. State of the Art

The interplay between topological persistence (TP) and networks is nowadays a wide and ramified research area; it dates back at least to [6], where the clique and neighborhood complexes were built on a time-varying network for application in statistical mechanics. TP was then applied to graph-derived simplicial complexes in several contexts: polymers, collaboration networks, various aspects of brain connections, social networks, language families, and many more.

Comparing large networks in search of a possible isomorphism is not only computationally unfeasible but also nonsensical; so a notion of *weak isomorphism* [14] and an interesting pseudo-metric on the space of all networks [15] were introduced. A natural strategy is the reduction to a set of invariants; along the same lines, the same authors applied persistent homology to the Dowker and clique complexes of a directed network [16,17]

and to its path homology [18]. The stability and convergence of the derived invariants are studied in a comprehensive paper: [19]. Two interesting papers on similar problems are: [20,21].

TP is used in [22] for assessing a sort of complexity of a neural network: each edge is given a weight derived from the activation function; then, the diagram of the persistent Betti number in degree zero is computed. Finally, the *p*-th "neural persistence" of the network is defined as the *p*-norm of the diagram. A "persistence interaction detection" framework is the core of [23]. This shows how TP can be of use in the analysis of neural networks; conversely, a neural network can be employed to directly produce the persistence image of a given picture [24].

The first paper we know, in which TP is part of a neural network, is [25], which adopts as the input layer a (vectorized) persistence diagram of the object to be classified. This is a strategy that was then refined by [26,27] and was followed in different forms by a large number of researchers. A similar technique, based on the "element specific persistent homology" is applied in a series of papers (starting from [8]) for the prediction of protein-ligand binding affinity; see the rich bibliography of [10]. In [28,29], the Wasserstein-1 distance between persistence diagrams contributes to the loss function of a deep neural network for 3D segmentation.

Neural networks that have graphs as input, benefit quite naturally from topological expedients for down-scaling: [30–33]. Coming closer to the subject of our research, a form of topological pooling based on persistent homology was defined in [34] for pose recognition; a *message reweighting graph convolution* is also based on TP in [35]. TP-based pooling is the subject of two well-structured papers: [36,37]. Still, these articles are based on the classical pipeline: constructing weighted simplicial complexes, getting a filtration, and computing persistent homology modules. Our approach differs in that we bypass the simplicial and homological passages, thanks to a generalization that produces persistence diagrams directly from graph-theoretical features.

## 3. Background

In the following paragraphs, we sketch and provide references for the essential mathematical constructions that motivate and inspire the idea of persistent-based layers: persistent homology, rank-based persistence, and steady persistent features.

### 3.1. Rank-Based Persistence

Persistent homology requires three main ingredients:
1. A filtered topological space;
2. The homology functor $S_k$ mapping topological spaces to finite vector spaces;
3. A notion of *rank*, e.g., the dimension for vector spaces or cardinality for sets [38].

In Figure 1, we show how considering a topological sphere $X \subset \mathbb{R}^3$ filtered by the height function

$$f : X \to \mathbb{R}$$
$$(x, y, z) \mapsto z$$

yields a persistence diagram whose points correspond to the maxima and minima of $X$ with respect to the vertical axis. We refer the reader to [39] for details on topological persistence and persistent homology.

We choose to frame our work in a more general context than homological persistence, namely *rank-based persistence* [11]. Although topological persistence and persistent homology have been extended in several ways, e.g., [40–44], rank-based persistence allows us to work directly in the category of the data of choice, rather than topological spaces. The authors assume an axiomatic standpoint based on the three building blocks of homological persistence mentioned above, and the authors generalize persistence to categories and functors other than topological spaces and homology. Importantly, under a few assumptions, persistence built in the rank-based framework still guarantees funda-

mental properties such as flexibility (dependence on the filtering function), stability [45], and robustness [46]. In this setting, we can work with data types such as images and time series, without intermediary topological constructions. We refer to [11] for details and provide a list of analogies between classical and rank-based persistence in Table 1.

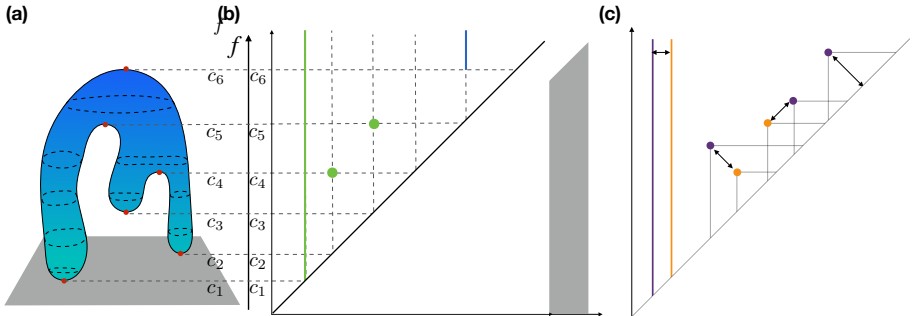

**Figure 1. Persistent homology**. Let us consider a topological space $X$ and a continuous function $f : X \to \mathbb{R}$. The (homological) critical values of $f$ induce a sub-level set filtration of $X$. (**a**) Critical values $C = \{c_1, \ldots, c_6\}$ of the height function on a topological sphere. Sublevels of the filtering function $f$—namely, $f^{-1}((-\infty, c])$, for every $c \in C$—yield a filtration of the topological sphere. (**b**) Changes in the number of generators of the $k$th homology groups along the filtration can be represented as a persistence diagram. 0-dimensional holes (connected components) are represented as green points. The void obtained at the last sublevel set gives rise to the blue line. (**c**) Persistence diagrams can be compared by computing an optimal matching of points. Unmatched points are associated with their projection on the diagonal.

**Table 1.** Analogy between the classical and rank-based persistence frameworks.

| Classical Framework | Categorical Framework |
| --- | --- |
| Topological spaces | Arbitrary source category **C** |
| Vector spaces | Regular target category **R** |
| Dimension | Rank function on **R** |
| Homology functor | Arbitrary functor from **C** to **R** |
| Filtration of topological spaces | $(\mathbb{R}, \leq)$-indexed diagram in **C** |

*3.2. Persistent Features*

In the spirit of the aforementioned generalization, ref. [12] (Section 2.2) introduces the concept of *steady persistent features* for weighted graphs.

A weighted graph is a pair $(G, f)$, where $G = (V, E)$ is a graph defined by a set of vertices $V$ and edges $E$, and $f$ is a function assigning (tuples of) real-valued weights to the edges of $G$, in symbols $f : e_i = [v_{i,1}, v_{i,2}] \mapsto w_i$. The weighting function naturally induces a sublevel set filtration

$$\varnothing \subseteq S_1 = f^{-1}((-\infty, w_1]) \subseteq \cdots \subseteq S_n = f^{-1}((-\infty, w_n]) = G.$$

For an intuition, see Figure 2a,b.

Let $S = 2^{V \cup E}$ be the set of all subsets of elements (vertices and edges) of $G$. Let $F$ be a graph-theoretical property, e.g., local degree prevalence, independence. The *persistent feature* $\mathcal{F} : S \to \{\text{true}, \text{false}\}$ associated with $F$ is a Boolean mapping returning true if the property $F$ holds for a certain subset $s \in S$ and false otherwise. Symmetrically to the topological persistence framework, we evaluate $\mathcal{F}$ for every subset of every $\{S_i\}_i$. See Figure 2b. Then, we compute the *steadiness* $\sigma$ of each subset $s$ along the filtration by counting subsequent sublevels such that $\mathcal{F}(s) = \text{true}$. As in [12], we refer to $\sigma((G, f, \mathcal{F}))$ as the steady persistence of the feature $\mathcal{F}$ on $(G, f)$. This construction yields a persistence diagram. See Figure 2c.

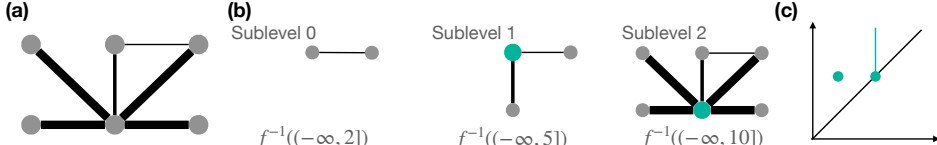

**Figure 2. Persistent features**. (**a**) A weighted graph $(G, f)$. The weight values—integers in the set $\{2, 5, 10\}$—are encoded as the thickness of the edges. (**b**) The sublevel set filtration induced by the weights. Teal vertices are the vertices in each sublevel with local degree prevalence. (**c**) The persistence diagram is obtained by considering the steadiness of each true subset along the filtration. One vertex realizes local degree prevalence at level 5 and stops being prevalent entering level 10. Another vertex is prevalent at level 10 and shall never stop being such because $f^{-1}((-\infty, 10]) = G$.

## 4. Persistence-Based Layers

In the following sections, we first build a persistence-based operator that can act as a filter on grayscale images (the operator can also be applied to RGB images treating each channel independently). This construction follows naturally from the definition of steady persistent function. Then, we discuss the main properties of such operators: locality and equivariance. Finally, we specialize our construction to operate as a pooling layer in a convolutional neural network.

### 4.1. Persistent Features as Equivariant Filters

Locality and equivariance are crucial features for convolutional neural networks, and in general for any group-equivariant model. Indeed, we have:

1. The intensity of a pixel in a grayscale image carries knowledge only when compared to neighboring pixels;
2. Identical configurations located in different regions of the image (translated) should be recognized as such by the model, as is the case for convolutional neural networks [47]. Mathematically, a function $f$ is equivariant with respect to the action of a group $G$ if $f(gx) = gf(x)$. See [4,9,48,49] for an overview on and examples of equivariant machine learning models

#### 4.1.1. Locality

In this setting, considering the notion of the persistent feature introduced in Section 3.2, we think about an image as a graph in which vertices are pixels and edges connecting adjacent pixels. Patches (or windows in a time series) of size $k$ around a pixel (point) correspond to $k$-distance neighborhoods of the vertex associated with such a pixel.

#### 4.1.2. Flexibility

Persistent features require a filtered space to be computed. Thus, after associating a graph to an image (or time series), we define the weighting function $f : S \to \mathbb{R}^n$, where $S$ is the set of all subsets of $V \cup E$ and $n \in \mathbb{N}$. Importantly, $f$ can carry additional information about the original data. For instance, when considering images, one can associate with each vertex the intensity value of its underlying pixel, and leverage this information to compute appropriate weights in a process reminiscent of message passing in graph neural networks [50]. The weighting function $f$ induces a sub-level set filtration of the graph associated with each pixel of the image. See Figure 3a,b for an example.

#### 4.1.3. Equivariance

Once the pair $(G, f)$ has been associated with our data, the proposed construction is naturally equivariant with respect to translation. Indeed, sublevel set filtrations and persistent features are totally determined by the weights and connectivity of the graphs at each filtration level. It is important to notice that the flexibility of the proposed solution makes it possible to further control the equivariance of the operator. For instance, weighting functions that only depend on pairwise intensity values will generate operators that not only

are equivariant with respect to translations, but also to isometries (translation, rotations, reflection, and scaling) of the original data.

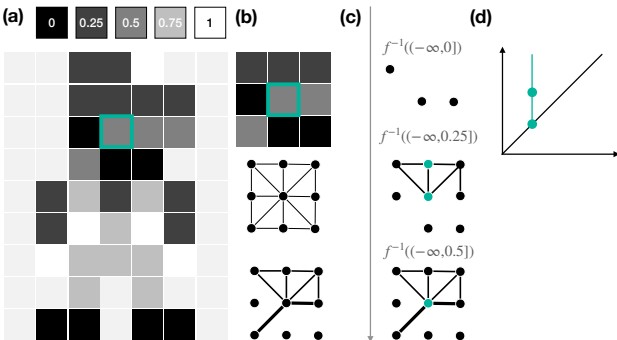

**Figure 3. Persistent features**. An image (**a**) and its interpretation as a weighted graph. (**b**) Given a focus pixel and its neighborhood (a $3 \times 3$ patch in the example), we map the image to a graph $G$ having as vertices the patch's pixels and edges connecting adjacent pixels. We weigh edges of $G$ by evaluating the function $f : \mathbb{R} \times \mathbb{R} \to \mathbb{R}$ mapping the intensity values of two vertices to the minimum intensity values of the respective pixels in the image. Edges with weight 0 are not shown. (**c**) The sublevel-set filtration induced by the weights. Teal vertices are the ones with a prevalent local degree. (**d**) The persistence diagram obtained considering the filtration/feature pair.

### 4.1.4. Parametrization

We can add parameters to the weighting function $f$ and the feature $\mathcal{F}$ to create operators endowed with more complex equivariance and learnable parameters. Let $S_t = \iota^{-1}((-\infty, t])$ be a sublevel of $G$ naturally induced by the intensity of pixels (edges are added when the vertices they connect are added). We define $\mathcal{G}_{m,n}^k(v, t)$ as the feature mapping $v$ to true if more than $m$ and less than $n$ pixels in the $k$-distance neighborhood of $v$ have an intensity less than $t$. Figure 4a shows how varying parameters $m$ and $n$ allow us to highlight radically different aspects of a binary image.

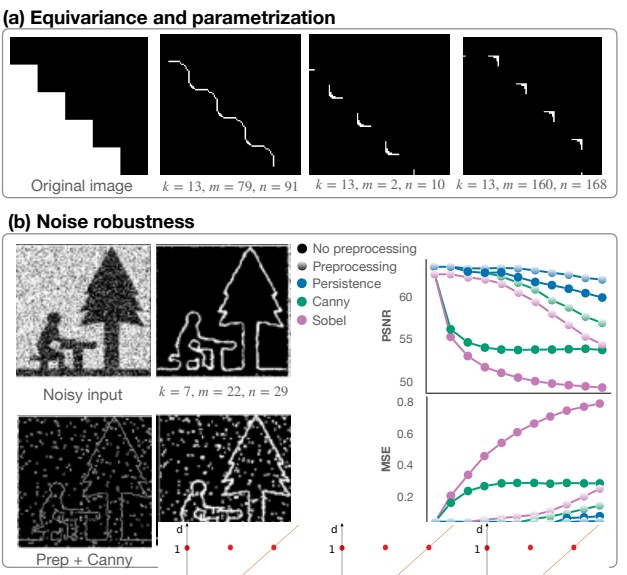

**Figure 4. Persistence-based filter**. (**a**) The persistence-based filter acts in an equivariant fashion (the same feature is recognized throughout the image) and parametrization changes lead to the detection of heterogeneous features. (**b**) We perform edge detection on a noisy input (40% of the pixels are salt and pepper noise). Performance is compared across the proposed persistence-based filter, Canny, and Sobel edge-detection algorithms with and without preprocessing (median filter). Results are showcased for different ratios of noisy pixels. On the right, we compare the Mean Square Error and Peak Signal to Noise Ratio across the considered edge-detection algorithms.

**Remark 1.** *The operator $\mathcal{G}$ defined above does not rely on the 2-dimensional structure of the selected patch. Thus, its steady persistence diagram is not only invariant with respect to translations, but also to permutations. This kind of equivariance meshes the standard convolutional equivariance with the fully-connected input/output representation typical of dense layers of an artificial neural network.*

4.1.5. Robustness

In the context of both topological and rank-based persistence, robustness means that small changes in the image do not give rise to significant perturbations of the corresponding persistence diagram. This concept is defined in [12] for persistent features and dubbed *balancedness*. In [51], it is formally shown that $\mathcal{G}$ is a *balanced feature* in the sense of [12]. Robustness to noise is showcased in Figure 4b. Additionally, and in line with the principles of functional data analysis [52], the proposed operator is adapted to work on continuous data and multiple resolutions: salient features are maintained across different resolutions, as shown in [53] in the context of persistent images; in computational experiments, the authors demonstrate how downsampling the persistence diagram does not affect the classification accuracy.

*4.2. A Steady, Persistent-Feature Layer*

The filter $\mathcal{G}$ and its steady persistence introduced in Section 4.1 is endowed with features inherited from its mathematical foundation, which make it suitable for tackling typical machine learning tasks:

1.  $\mathcal{G}$ enhances the signal in correspondence of abrupt changes (a max-pooling filter could be blind to such features);
2.  The steady persistence $\sigma(\mathcal{G})$ yields invariant representations of the input with respect to the group of isometry;
3.  Salt-and-pepper noise does not impair the quality of detected features.

These properties motivate implementing and testing $\sigma(\mathcal{G})$ as a complement to pooling and convolutional operators in standard artificial neural networks. In the following paragraphs, we discuss the implementation of a pooling layer with learnable parameters based on persistence features. The same operator can be easily adapted to work as a convolution-like layer.

4.2.1. Persistence-Based Pooling

We consider an image $I$ and split it in a collection of patches $\{P(h,w)_i\}$ of size $(h,w) \in \mathbb{N}^2$. For every pixel $p \in P_i$, we compute $\sigma(\mathcal{G}_{m,n}^k)(p)$ for some fixed values of $m, n$ and $k$ (padding is added if needed). This procedure, which computationally boils down to sorting and slicing operations, yields a persistence diagram with a point per pixel in $P_i$. Indeed, we compute the operator $\mathcal{G}$ for every $P_i$, padding the patch whenever necessary. Symmetrically to the classical max-pooling operator, the maximum persistence—i.e., the distance from the diagonal—shall determine the value to be associated with the entire patch and its downsampling. As an example, see Figure 5a,b, and the top row of (c).

4.2.2. Parametrization

Alternatively, and following the idea of learnable pooling operators, e.g., [13], we propose to learn weights $\Lambda$ to modulate the contribution of the non-zero component of the persistence diagram, as depicted in Figure 5c. Because the persistence value associated with each pixel is continuous and $\mathcal{G}$ is balanced, it is possible to learn such weights through standard backpropagation.

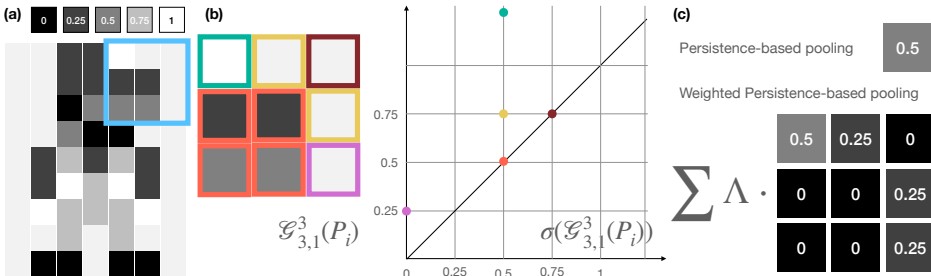

**Figure 5. Persistence-based Pooling**. (**a**) We extract patches of fixed size from the input image. (**b**) The steady persistence operator $\sigma(\mathcal{G})$ yields a persistence diagram associating a persistence value—namely, the distance from the diagonal of the points of the persistence diagram on the right of the panel—with each of the pixels of the considered patch. (**c**) We associate to each patch either the maximum persistence or a weighted sum of the persistence values associated with each pixel. In the latter case, weights are learned through gradient descent.

## 5. Computational Experiments

We assess the performance of the suggested pooling layer, embedding it into two neural architectures. First, we compare the performance of the persistence-based pooling with some classical pooling implementations. Then, we compute saliency maps (projections of the network's gradients onto the input images) to highlight qualitative differences across the considered pooling layers.

### 5.1. Datasets

We perform computational experiments on the MNIST [54], Fashion-MNIST [55], and CIFAR-10 [56] datasets. The MNIST and Fashion-MNIST datasets are composed of grayscale images labeled according to ten classes of hand-written digits and fashion articles, respectively. Both datasets are composed of 60,000, 28 by 28 pixels, black and white images for training, and 10,000 for testing. The CIFAR-10 dataset is composed of 50,000, 32 pixels by 32 pixels, RGB images for training that belong to ten classes: airplane, automobile, bird, cat, deer, dog, frog, horse, ship, and truck. The test is performed on 10,000 additional labeled images.

### 5.2. Architectures

We test the proposed layer via two neural network architectures depicted in Figure 6. We designed the first architecture to provide the simplest configuration, in which different pooling operators could be compared during a supervised task where parameters are learned via gradient propagation. The second architecture is more akin to the standard convolutional neural network topology in a simple image-classification task; we alternate the convolutional and pooling layers before a dense classifier is provided.

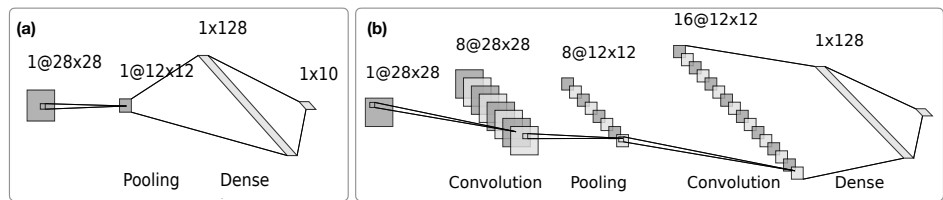

**Figure 6. Benchmark architectures.** The two architectures used in computational experiments. (**a**) A toy neural network where a pooling operator is applied directly to input images. The downsampled output of the pooling layer is passed to a dense classifier. In other words, first, the pooled output is flattened, then its dimensionality is reduced to match the number of classes to be predicted by the network. (**b**) As common practice for CNNs, this architecture alternates convolutional and pooling layers before the dense classifier.

Training

We use sparse categorical cross-entropy as a loss function [57]. Optimization is conducted in batches of 32 images through the ADAM optimizer [58] with a learning rate of $3 \times 10^{-4}$. We train each model for 100 epochs and make use of early stopping [59] with parameters min_delta = 0 and patience = 3.

### 5.3. Results

We tested the architectures presented in Section 5.2 on image classification on the MNIST, Fashion-MNIST, and CIFAR-10 datasets. We adapted the persistence-based and convolutional layers to work with the RGB images of CIFAR-10. Specifically, the persistence-based pooling layer treats channels independently; thus, multiple channels are processed in parallel, ignoring their interactions.

### 5.3.1. Performance

We compared the performance of several pooling layers, namely:

- Max-pooling;
- Persistence-based-pooling with learnable parameters (ours);
- LEAP [13].

Additionally, we considered a combination between the learnable persistence-based pooling and max-pooling layers. The two pooling approaches are combined by adding the value obtained through max-pooling on a specific patch to the weighted sum defined in Section 4.2 and depicted in Figure 5c. In this way, the weights learned during training combine and modulate the contribution of persistent features detected in each patch and the maximum intensity value of the pixels therein.

We quantify the performance of each network through a standard accuracy metric. We list the results in Table 2. The proposed layers, outperform the max-pooling and LEAP on both datasets and architectures. The combination of max- and persistence-based-pooling realizes the maximum performance in three out of six experiments. We believe this result points to the complementarity of the two approaches.

**Table 2.** The accuracy realized by the selected architectures when endowed with ours and state-of-the-art pooling layers.

| Architecture-Dataset | Max | LEAP | PL (Ours) | PML (Ours) |
|---|---|---|---|---|
| (a)—MNIST | 0.8905 | 0.9238 | **0.9472** | 0.9087 |
| (b)—MNIST | 0.9886 | 0.9848 | **0.9908** | 0.9880 |
| (a)—FMNIST | 0.7978 | 0.8385 | 0.8424 | **0.8435** |
| (b)—FMNIST | 0.8845 | 0.8776 | 0.8930 | **0.8985** |
| (a)—CIFAR-10 | 0.3226 | 0.3291 | **0.4185** | 0.3869 |
| (b)—CIFAR-10 | 0.6145 | 0.5217 | 0.6355 | **0.6499** |

### 5.3.2. Qualitative Comparison of Salient Features

Pooling operators aim to select salient features of their input. Grad-CAM heatmaps [60] are a visualization tool for highlighting regions of the input that contributed most to the model's prediction. Figure 7 showcases Grad-CAM heatmaps obtained through the considered architectures on random input samples. As expected, the persistence-based pooling and its combination with max-pooling yield similar yet distinct heatmaps. In particular, although non-topological, the persistence-based pooling seems to capture geometrical and topological features of the images, such as corners and regions that would cause the foreground image to disconnect. See Figure 7a,b, respectively.

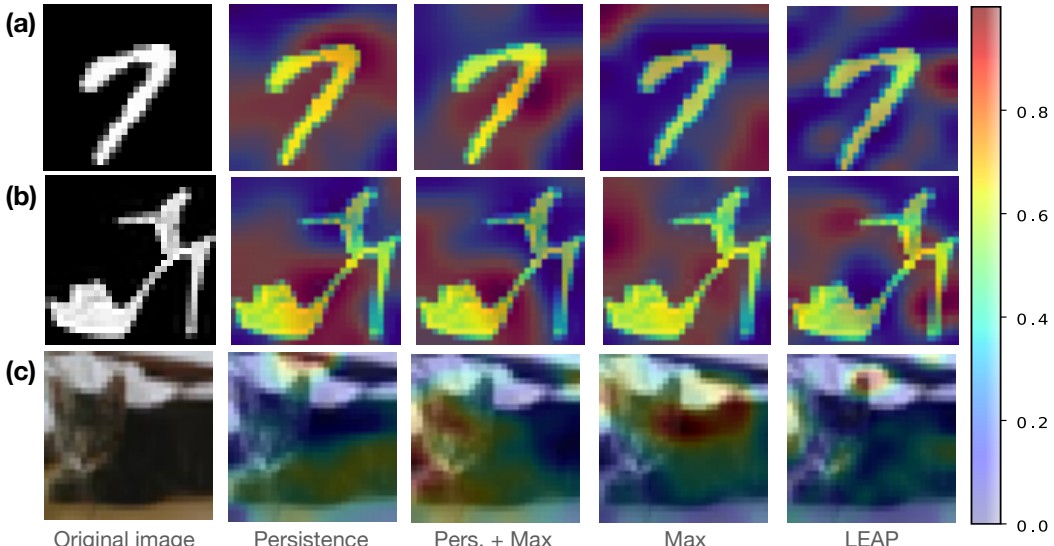

**Figure 7.** Grad-CAM heatmaps show pixel-wise saliency of features of an image with respect to the neural network's gradients. Each row provides an example from a different dataset: MNIST (**a**), Fashion-MNIST (**b**), and CIFAR-10 (**c**).

## 6. Conclusions

At the crossroad between artificial intelligence and generalizations of topological data analysis, we propose original constructions of neural network layers, taking advantage of the formalism of rank-based persistence. Building on persistent features, we define a convolution-like operator that can be tailored to specific tasks by imposing equivariance through simple invariants. Such invariants can be defined easily by considering features of the data at hand rather than mapping data to the category of topological spaces. Thanks to its mathematical foundation relying on steady persistence and inherited by persistent homology, the proposed operators enjoy mathematically guaranteed properties such as noise robustness and stability to perturbations. We showcase these properties by explicitly defining an image filter operator, relying on steady persistence and testing it in an edge-detection task on noisy images. We compare the performance of the proposed filter against state-of-the-art edge detectors, namely Canny and Sobel operators, comparing the mean squared error and peak signal-to-noise ratio achieved by the three methods on images affected by the salt and pepper noise. In our tests, the proposed operator outperforms competitors in detecting edges and image restoration (noise removal).

Locality, equivariance, and robustness to noise make the proposed class of operators a plausible complement to existing neural layers. Indeed, neural network layers are designed to be as agnostic as possible to the intrinsic properties of their input data. These features make them incredibly general and, at the same time, challenging to constrain to specific data features. This generality induces problematic behaviors and hinders the intelligibility of neural networks' learning patterns [61–63]. The proposed persistence-based layers, allowing the user to select relevant (persistent) features *a priori*, pave the road to the design and implementation of hybrid strategies leveraging the data-driven, learnable nature of artificial neural networks and the flexibility of the persistence-based approach.

We devise and implement a special class of the persistent-features-based layer called persistent-based pooling. The proposed layer uses the previously defined edge detection operator to act as a pooling layer in convolutional neural networks. A natural choice of operation for persistence-based pooling is to consider the maximum persistence realized in the diagram associated with each input image patch. We implement an alternative parametrized version that is equipped with learnable weights and easily combinable with standard max-pooling. We test this learnable layer version in image classification on three benchmark datasets. We compare the classification accuracy achieved by our layers

with max-pooling and LEAP and embed the pooling layer into two neural architectures (dense and convolutional, respectively). Persistence-based pooling realizes a better performance than its competitors across architectures and datasets. Finally, utilizing Grad-CAM heatmaps, we visualize salient features of input samples to provide a qualitative comparison across the considered pooling layers. There, the persistence-based pooling seems to retrieve relevant geometrical properties of the images.

This approach meshes well with the framework developed in [64], where inputs, outputs, and weights of a neural network layer are expressed as functions on smooth or discrete spaces. There, we demonstrated how several classes of *linear* neural network layers can be expressed as a combination of the function pullback (from a smaller to a larger space), pointwise multiplication of functions, and integration along fibers. We believe that, by combining the two approaches, it will be possible to define general parametric nonlinear layers, where the architecture is defined through the *parametric spans* introduced in [64], whereas the geometry-aware nonlinear computation descends from the methods developed here.

**Author Contributions:** Conceptualization, M.G.B., M.F., A.M. and P.V.; methodology, M.G.B., M.F. and P.V.; software, validation, data curation, M.G.B. and A.M.; writing—original draft preparation, M.G.B.; writing—review and editing, M.G.B., M.F., A.M. and P.V.; supervision, M.F. All authors have read and agreed to the published version of the manuscript.

**Funding:** This research received no external funding.

**Data Availability Statement:** No datasets were generated during the current study. The datasets analyzed are available at MNIST, http://yann.lecun.com/exdb/mnist/, Fashion-MNIST, https://github.com/zalandoresearch/fashion-mnist, CIFAR-10, https://www.cs.toronto.edu/kriz/cifar.html, Code available at https://gitlab.com/alex920929/thesis-code (all accessed on 27 December 2022).

**Acknowledgments:** We thank Maurizio Sanarico for the useful discussions. Work performed under the auspices of INdAM-GNSAGA.

**Conflicts of Interest:** The authors declare no conflict of interest.

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
