# Peer review of "Generalized Persistence for Equivariant Operators in Machine Learning"

_make, doi:10.3390/make5020021_

Round 1
Reviewer 1 Report
This paper introduces persistence-based neural network layers, which attempt to allow users to inject knowledge about symmetries (equivariance) respected by the data.
Strength
The core idea is a good one, and it leads to promising results.
Weakness
A few experiments are performed to demonstrate that the idea works, and there is little explanation of the experimental setting and results.
There is a lack of clarity and justification in the theoretical approach
The paper is poorly organized and needs to be better checked before submission. It contains a lot of grammar errors and is not well-written.
Author Response
A few experiments are performed to demonstrate that the idea works, and there is little explanation of the experimental setting and results.
Greater care has been given to the explanation of the experimental setting and results: lines 266-268, 274-275, 281-287, 293-300.
There is a lack of clarity and justification in the theoretical approach
We have tried to improve this aspect: lines 38-48, 162-166, 208-213, Caption of Fig. 5.
The paper is poorly organized and needs to be better checked before submission.
The initial part of the Introduction (lines 10-36) and the Conclusions section (lines 302-339) have been completely rewritten, as well as the initial parts of the single sections (lines 110-112, 151-154, 247-248).
We have even changed the title in order to make it more adherent to the content of the paper.
It contains a lot of grammar errors and is not well-written.
We have (hopefully) amended all errors and made the text clearer and smoother.
Thank you very much.
Reviewer 2 Report
After reading the article "Persistence-based operators in machine learning" (Authors: Mattia G. Bergomi, Massimo Ferri, Alessandro Mella, Pietro Vertechi), the following remarks should be mentioned:
1. The title of the article is missing.
2. Subheadings "Aim.", "Contribution.", etc. must not contain a dot at the end.
3. Strange formatting on p. 3, lines 65-68.
4. Strange list formatting on p. 5, lines 101-105.
5. Strange list formatting on p. 7, lines 148-153.
6. Incorrect referencing on p. 8, line 182.
7. Too much self-citations is bad style. One gets the impression that there are practically no works on this topic, except for articles by Bergomi M.G. The Introduction should give a broader view of the problem and publications in this area.
Summary:
The work makes a good impression. The comments are mostly about formatting and minor. In general, it is adequately structured, the problem and its solution are described in sufficient detail. There is both a mathematical basis and an experimental part. I would like to note the detailed illustrations.
I will recommend the article for publication after correcting minor comments and improving the Introduction section.
Author Response
1. The title of the article is missing.
2. Subheadings "Aim.", "Contribution.", etc. must not contain a dot at the end.
3. Strange formatting on p. 3, lines 65-68.
4. Strange list formatting on p. 5, lines 101-105.
5. Strange list formatting on p. 7, lines 148-153.
6. Incorrect referencing on p. 8, line 182.
All of these are my mistakes (M.F.); I apologize. They should have been amended now.
We have also changed the title to make it more adherent to the content of the paper.
7. Too much self-citations is bad style. One gets the impression that there are practically no works on this topic, except for articles by Bergomi M.G. The Introduction should give a broader view of the problem and publications in this area.
The initial part of the Introduction (lines 10-48) has been re-written, and above all the section State of the art (lines 72-108) has been added.
Thank you very much.
Reviewer 3 Report
The paper explores topological analysis (topological persistence, TP) in the design of NN layers. The proposed persistence based pooling layers are experimented with two architectures (Dense and Convolutional) on traditional data sets. Results indicates that the proposed layers outperform max-pooling and LEAP, and the combination max- and persistence-based-pooling achieves maximum performance
The article addresses a very current and important topic in the area. The text is well written, the mathematical formalization is consistent, and the general methodology is sound. Pictures have high quality and help in the understanding of the theory and design of the solutions. Results are promising and described in sufficient detail in the text.
I have no corrections or further experiments to suggest (given the short review cycle).